# Modification of Polymeric Surfaces with Ultrashort Laser Pulses for the Selective Deposition of Homogeneous Metallic Conductive Layers

**DOI:** 10.3390/ma15196572

**Published:** 2022-09-22

**Authors:** Michael Seiler, Andreas Knauft, Jann Jelto Gruben, Samson Frank, Andrea Barz, Jens Bliedtner, Andrés Fabián Lasagni

**Affiliations:** 1Fachbereich SciTec, Ernst-Abbe-Hochschule Jena, Carl-Zeiss-Promenade 2, 07745 Jena, Germany; 2Institut für Fertigungstechnik, Technische Universität Dresden, George-Baehr-Str. 3c, 01069 Dresden, Germany; 3Fraunhofer Institut für Werkstoff und Strahltechnik IWS, Winterbergstr. 28, 01277 Dresden, Germany

**Keywords:** polybutylene terephthalate, polyamide, picosecond laser, surface activation, palladium distribution, selective metallization

## Abstract

In recent years, the demand for highly integrated and lightweight components has been rising sharply, especially in plastics processing. One strategy for weight-saving solutions is the development of conductive tracks and layouts directly on the polymer housing parts in order to be able to dispense with the system integration of additional printed circuit boards (PCB). This can be conducted very advantageously and flexibly with laser-based processes for functionalizing polymer surfaces. In this work, a three-step laser-based process for subsequent selective metallization is presented. Conventional injection molded components without special additives serve as the initial substrate. The Laser-Based Selective Activation (LSA) uses picosecond laser pulses to activate the plastic surface to subsequently deposit palladium. The focus is on determining the amount of deposited palladium in correlation to the laser and scan parameters. For the first time, the dependence of the metallization result on the accumulated laser fluence (*F_acc_*) is described. The treated polymer parts are characterized using optical and scanning electron microscopy as well as a contact-type profilometer.

## 1. Introduction

The development of smart, lightweight, and compact devices has experienced significant growth in recent years. The automotive industry, the lighting industry, and the media sector have great potential for these trends due to the increasing demand for innovations and products, such as molded interconnected devices (MID) or smart wearables [1,2,3].

A key target of development for solving these requirements is the flexible production of electrically conductive tracks for building individual electronic circuits and systems. Industrial solutions offer various suitable technologies for structuring and metallizing polymeric and dielectric materials. The main aim of these methods is to change the conductivity of the insulating surface properties. Often, there is a requirement to achieve high contrast between the conductive structure to be created, with very low electrical resistance, and the surrounding unaffected area, with very high electrical resistance.

Photolithographic processes are commonly applied for two-dimensional structuring tasks. For three-dimensional applications, on the other hand, there are a number of laser-based technologies that can be implemented. These include Laser Metal Deposition (LMD) [4,5,6], Laser Direct Writing (LDW) [7,8,9,10], Laser-Induced Forward Transfer (LIFT) [11,12], Laser-Induced Selective Activation (LISA) [13,14], Laser Direct Structuring (LDS) [15,16,17], and Selective Surface Activation Induced by Laser (SSAIL) [18,19,20,21,22,23]. Table 1 summarizes selected methods with their individual characteristics.

The literature labeled * in Table 1 is the most relevant for comparison with our own work and therefore needs to be examined in detail. In principle, the application of a laser process offers many advantages, such as localized energy input, the production of small structures, good automation capability, and the possibility of treating very different materials. In all cases, laser radiation as a tool is used to initiate or support the activation process on the technical surfaces. For the presented methods, the typical laser wavelengths range from ultraviolet (UV) to infrared (IR). Different physical interaction mechanisms occur due to the different wavelengths and pulse parameters. On the one hand, the surface is roughened, and on the other hand, the polymer bonds are modified. The highlighted methods in Table 1 meet both requirements. After that, the electroless deposition of metals (Cu, Ni, Au) is generally applied, which is based on a nucleation step with a seed material (e.g., Pd, Ag).

The authors Shafeev et al. and Schrott et al. presented laser-based methods such as LMD for the metallization of dielectrics. Different materials (see Table 1) were irradiated with excimer laser radiation while immersed in various concentrations of Pd solution [4,5,6,24,25]. The authors have demonstrated a laser-based decomposition of palladium acetylacetonate dissolved in dimethylformamide to produce catalytic Pd particles for subsequent electroless plating [36]. LMD was investigated under air and resulted in catalytically active surfaces. An advantage of using nanosecond excimer lasers is ablation by high photon energy (≈5 eV) in the UV, which leads to the breaking of polymer chains within a few nanoseconds [37].

The methods of LDW processes and Laser-induced Chemical Liquid Deposition (LCLD) were investigated by Kordas et al. Thin films of Pd and Cu could be deposited on the polymers PI and PET as liquid precursors with XeCl (λ = 308 nm) and KrF (λ = 248 nm) lasers or by a photothermal reaction with a cw Ar^+^ laser (λ = 488 nm) [8,9,26,27,28].

Already in 1970, the LIFT technique was used for the activation of material through an air gap, as mentioned in the work of Levene et al. In 1986, the process was also applied by Bohandy et al. for the transfer of metals. A thin film on the back of a substrate, such as glass, was heated with a laser source. After reaching the boiling point, the metal was transferred to the target by the vapor-driven energy. The disadvantages of this method are the weak adhesion between the donor and target substrates and the lack of transferability to 3D substrates [11,29,30,38].

LISA is a promising method for the application of 3D-MID. The principle of this technology is based on laser treatment in distilled water, an activation bath, and metallization by a coating process. In [14], two wavelengths were used for treating polycarbonate (PC) surfaces without metallic fillers. Increasing scan repetitions leads to the carbonization of the polymers. In the mentioned research, a first correlation was introduced between the roughness parameters and the laser parameters. Due to the burning and carbonization tendency of the material utilized, extensive further quantification measures were required.

In the case of the LDS process, metallic seed particles or fillers such as carbon nanotubes (CNTs) are required in the polymer to achieve metallization [16,35,39]. For this purpose, the additives are mixed into the injection molding compound. However, the currently available range of materials is limited. For example, two variants of polybutylene terephthalate (PBT) polymer named Pocan and Vestodur were processed by Islam et al. which did not show sufficient metallization properties [40]. Various fillers (metal oxide composite of copper-chromium oxide) were applied separately or in combination for ABS and PDMS in Zhang et al. [31,32]. Currently, the LDS process is the industrially established method for fabricating 3D-MIDs and other electronic devices. The disadvantages of the method are the increased material costs due to the inorganic dopants, the sometimes heterogeneous distribution of additives in the molding, and the occurrence of impurities after injection molding, which are not suitable for some laser applications [38].

In addition to the LDS method, SSAIL was also utilized for selective metal deposition. Here, nano- and picoseconds in the IR and VIS range are used to modify the surface under atmospheric conditions. Silver nitrate solutions are used as a catalyst for subsequent electroless copper deposition. In addition to the polymers studied, a dependence of the measured conductivity of the resulting copper films on the applied laser fluence was also demonstrated. The nanosecond pulses used do not lead to copper plating. SSAIL is not yet industrially established. Demonstrators for three-dimensional applications have already been shown. The application of laser pulses in the range of several picoseconds was reported by Ratautas et al. and Camargo et al. The laser sources allow higher precision, higher contrast, and negligible heat input during the patterning of the polymeric surface [16,18,19,20,21,22,23,33,34,35,41]. However, to the best of the authors’ knowledge, the parameter accumulated fluence has not been investigated, and thus, no correlation between the activation of the surface and copper plating has been established. Furthermore, the material PBT is not found in the literature for SSAIL and LISA. Therefore, the existing literature does not provide a comprehensive model for understanding laser-based surface activation on PBT. It is currently not known whether the influence of roughness or chemical activation on germination and later copper plating.

This study describes the utilization of the Laser-based Selective Activation (LSA) method to generate conductive structures on PBT and polyamide (PA6) polymer substrates under atmospheric conditions [42]. A major goal of the investigation is to demonstrate the selective metallization on the above-mentioned polymeric surfaces. The three serial process steps, activation, nucleation, and metallization, are evaluated. With the use of analytical methods, the apparent amount of Pd on the surface of the polymers is quantified as a function of the laser and scan parameters. Methods of image analysis are used to determine the dependence of the selective metallization on the accumulated laser fluence. As a result of these investigations, process limits and a process window for the investigated polymers are shown.

## 2. Materials and Methods

### 2.1. Materials

As substrate materials, injection molded polymers made of PBT and PA6 were selected. The trade name of PBT is Arnite^®^ green from DSM (Heerlen, The Netherlands), and that of PA6 is Durethan, naturally transparent from LANXESS (Cologne, Germany). The polymers were unreinforced, but PBT has flame-retardant additives. The substrates were injection molded as square-shaped sheets (60 × 60 × 3 mm^3^) (provided by 3D Schilling GmbH, Sondershausen, Germany). In addition to the significant material parameters summarized in Table 2, the roughness of the surfaces is of interest for further process steps. The measured surface roughness (*R*q) was 30 nm for PBT and 60 nm for PA6 after the molding step. These very low roughness values, first of all, fundamentally ensure that palladium enrichment cannot take place on the substrate material. The specimens were stored under normal atmospheric conditions. The average temperature and relative humidity correspond to 24 °C and 46%, respectively.

Previous experimental investigations have shown that polymers with double bonds are suitable for the activation process and particularly reactive for downstream palladium nucleation [41]. They are used primarily in the automotive and household appliance industries and are, therefore, of particular interest for MID solutions.

### 2.2. Process Chain for Fabrication of Conductive Traces

The process chain used in this work to fabricate the conductive traces consists of three serial steps shown in Figure 1: (a) laser activation, (b) infection bath, and (c) selective metallization. A major objective of the investigation was the formation of homogeneous electrically conductive tracks (Figure 1d) based on the patented process LSA.

For laser activation (Figure 1a) of the surface, an experimental setup with a picosecond laser (TruMicro 5050, Trumpf GmbH & Co. KG, Ditzingen, Germany) was used to activate the substrate. The system emits a laser beam with a wavelength of 1030 nm and a maximum pulse repetition rate of 200 kHz. The pulse duration is constant at 7 ps. The laser beam was deflected by a dynamic galvo scanner (hurryscan II 14, Scanlab GmbH, Puchheim, Germany). An f-theta objective (Linos Ronar, Qioptiq Photonics GmbH & Co. KG, Göttingen, Germany) with a focal length of 100 mm was employed. This resulted in a focal diameter (*d_f_*) of 30 µm and a maximum scan field of 60 mm × 60 mm.

To investigate the process, a matrix of 5 × 5 fields was created on each sample under atmospheric conditions and at room temperature, as shown in Figure 2. The individual fields cover an area of 8 mm × 8 mm. The scan path (*y*-direction) was unidirectional, and the number of passes was set to one. For all experiments, the pulse repetition rate was set to 25 kHz.

To achieve laser fluences (*F*) from 7 to 35 J∙cm^−2^, the pulse energy varied between 25 and 125 µJ. The focal diameter *d*_F_ is used for the calculation of the circular area *A*. Equation (1) can be used to determine the fluence *F*.
*F = 2E_P_/A*(1)

The parameter *v_s_* corresponds to the scanning velocity and is used to vary the pulse distance *d_P_* (Figure 2). The pulse distance *d_P_* can be calculated using Equation (2)
*d_P_ = v_s_/RR*(2)
where *v*_s_ is the scanning speed and *RR* is the repetition rate of the laser source. The pulse (*d_P_*) and line distances (*d_L_*) were varied in the experiments from 10 to 80 µm in order to find the process limits for a homogeneous copper coating deposition.

The infection bath step (Figure 1b) consisted of treatment in PdCl_2_ solution with a concentration of 1 g∙L^−1^ PdCl_2_ (99.9%, Carbolution Chemicals GmbH, St. Ingbert, Germany). The laser-treated samples were directly immersed in the solution for 5 min at 70 °C. Subsequently, the samples were rinsed twice with distilled water.

The final step of the procedure consists of selective metallization of the substrate (Figure 1c) by electroless copper deposition. In this case, a commercial copper bath (Enplate LDS Cu-100, MacDermid Enthone, Langenfeld, Germany) was used. The samples were immersed for 5 min at a controlled temperature of 45 °C. They were then rinsed in distilled water and air dried. The effect on the variation of chemical baths has been published elsewhere [41].

### 2.3. Surface Analysis

A digital microscope (VHX-2000, Keyence Deutschland GmbH, Neu-Isenburg, Germany) with a ×200 to ×1000 zoom lens was used to examine the surface topography of the microstructured samples and the subsequent copper layers. Surface roughness was determined using a contact-type profilometer with an accuracy of 7 nm (stylus instrument, Talysurf i-Series 5, AMETEK GmbH, Weiterstadt, Germany).

Further evaluation of the sample surfaces was performed using a scanning electron microscope (SEM) (Evo Ma10, Carl Zeiss AG, Oberkochen, Germany). To determine the Pd content on the different surfaces, an energy dispersive X-ray (EDX) detector was used at 20 keV and a probe current of 1.5 nA. Areas of 300 × 400 µm^2^ were analyzed. The samples were coated with a carbon layer before the examination.

### 2.4. Evaluation of the Copper Distribution on the Laser-Treated Surfaces

The main objective of this work is to determine a process window for the activation step previous to the deposition of copper layers, obtaining a homogeneous layer. To ensure objective and quantitative results, an evaluation tool was developed using Matlab, which consists of the evaluation of images of the coated surface taken with optical microscopy. The following steps were applied:i.Characterization of the reference geometry;ii.The setting of fixation points;iii.Geometric transformation;iv.Grayscale scaling;v.Binarization and, finally, analysis of the copper distribution per field

A subsequent geometric transformation was used to clean up deviations and alignment errors from the camera image. Subsequently, the RGB images were decomposed into individual color channels. For binarization, only the red and blue color channels were used in order to achieve optimal threshold sensitivity for copper. The thresholds for global binarization were calculated by the method of Otsu [40]. The copper fraction was determined from the ratio of binarized area per field to the total field size. As a result of the image processing algorithm, a map was created showing the copper distribution as well as the metallized area as a percentage

## 3. Results and Discussion

Selected results of the three process stages are presented in this section. The analytical evaluation of the partial results represents the prerequisite for the complex understanding of selective metallization. The functional relationships between the laser fluence, scan parameters, and surface roughness are discussed. Similarly, the influence of these parameters on the achievable palladium concentration and distribution after the nucleation step is determined analytically. In the last process step, the functional relationship between the accumulated fluence and the resulting process windows for optimal deposition of copper for the two materials is discussed.

### 3.1. Polymer Surface after Laser Activation

By means of an experimental program, the process window was designed that includes different parameters such as laser fluence as well as line and pulse spacing in order to investigate their influence on the activation result. For these parameter combinations, a total of five samples for each material with 25 fields per sample result according to the selected experimental program. A dimension of 8 × 8 mm^2^ was selected for the field size. The aim of these investigations was the experimental determination of parameters that allow a homogeneous activation structure. The evaluation criteria for homogeneity of the laser-treated surface are the surface roughness of the activated area, the color change, and the contrast at the transition of the untreated surface to the activation area.

Figure 3a,c shows the produced color change after laser activation for both PBT and PA materials in the 5 × 5 fabricated matrix. The line spacing was 10 µm for PBT and 80 µm for PA. The two polymers are subject to two different reasons for this color change. For the PBT material (Figure 3a), a color change from green (initial state) to gray/dark gray (after laser activation) can be observed in all tests performed. In particular, the laser activation process performed using high laser fluence values, and low pulse spacing leads to darker colors, as can be seen in the top line of Figure 3a. The carbonization of the material is characterized by the local ablation in which the additives in the polymer matrix form a carbon black film [37]. This chemical-physical interaction has also been observed for other polymers such as PET, PS, PC, and PVC and depends on the fillers, stabilizers, and other particles in the matrix [45,46,47].

In contrast, a different behavior was observed in PA6 (Figure 3c). In this case, the color change results from foaming, where the polymer matrix scatters visible light [37].

In addition to the color appearance, Figure 3b,d also shows the formation of characteristic patterns on the surface and process boundaries that are exemplary for the investigations of both materials. Figure 3a shows the process boundary with red dashed frames. This initially represents the upper process limit for laser activation, where the formation of melt and the deterioration of the material begins. In contrast to PBT, Figure 3c shows a different process boundary for PA, which leads to an inhomogeneous color change, as can be seen in column 1 for a laser fluence of 7 J∙cm^−2^. The homogeneous discoloration of the fields indicates a modification threshold, as shown in Figure 3c (red dashed frame). If the fluence of 7 J∙cm^−2^ is exceeded, a constant white/greyish color change occurs. Up to a line distance of *d_P_* 40 µm, a homogeneous surface can be seen (Figure 3b, 7 J∙cm^−2^). Additionally, the parameter laser fluence can lead to a different overlap and, thus, a modified surface topography. A rising laser fluence increases, a larger spot at the samples is ablated, and thus the pulse overlap increases (Figure 3b, 35 J∙cm^−2^) [48]. If one of the scanning parameters exceeds this limit towards higher values, the activation process leads to ablation with a grooved structure or round craters (Figure 3d). In summary, it can be concluded that the resulting surface structures of the conducted experiments can be divided into three principal classes: (1) homogeneous structure, (2) linear grooves in *x*- or *y*-direction, and (3) round crater structures. These phenomena are also reflected in the roughness measurements discussed below.

In Figure 4, the squared mean roughness *Rq* of the different laser-treated areas of the polymeric surfaces is plotted as a function of both the laser fluence and pulse distance. Due to the chosen line and pulse distances, the morphology of the surface depends on the measurement direction (*y*- with scan direction and *x*- against scan direction). In order to explain the change in the laser-activated areas, only the measured values corresponding to the height profiles perpendicular to the scan direction are presented (see Figure 2).

As can be seen in Figure 4a,c, the measured *Rq* values for PBT and PA increase with increasing laser fluence. Differently, they decrease with pulse distances from 10 µm to 80 µm, as shown in Figure 4b,d. In the last case, due to the smaller pulse overlap, the laser-activated zone has fewer pulses per area, and thus *Rq* decreases. In particular, for pulse distances greater than or equal to 40 µm, the roughness of the laser-activated area is almost equal to the initial value for PA (Figure 4d). It has to be mentioned that the roughness values for almost all used process conditions were similar independently of the scan direction (not shown), with the exception of the samples treated with *d*_L_ of 80 µm.

The highest achieved roughness for both polymers was obtained for *d*_P_ = 10 µm, which is shown in Figure 4b,d. For PBT material, the highest roughness occurs at high fluences and small pulse spacings, which can be explained by the start of melting (Figure 3a red dashed frame). The *Rq* value is between 4.51 and 5.06 µm at a pulse spacing of 10 µm and a fluence of 35 J∙cm^−2^. In the first column, at a fluence of 7 J/cm^2^, inhomogeneous roughness results, leading to an increased Rq of 2.97 µm. In general, the resulting roughness *Rq* is higher for the PBT material than for PA. For small pulse distances below 40 µm and small laser fluences, homogeneous surfaces with low *Rq* values are obtained. At very large pulse and line distances over 60 µm, no pulse overlapping occurs when the pulse distance exceeds the focal diameter (for example, see Figure 3d at *d*_L_ = 80 µm. This leads to the formation of grooves and craters, which can result in different roughness values in both directions.

### 3.2. Analysis of the Pd Content and Distribution on the Polymer Surface

A crucial requirement for copper deposition using chemical plating is the presence of a certain concentration of palladium which is nucleated after the chemical baths using the PdCl_2_ solution [41] and can be controlled by the laser treatment. In consequence, a quantitative evaluation of the distribution and amount of Pd as a function of the process parameters is important for understanding the metallization process.

Figure 5 shows representative SEM and EDX images of laser-treated PBT samples, indicating the Pd content measured with EDX and topography measured with SEM for different scan and laser parameters. First, a periodic structure of spots can be seen in Figure 5a, in which the laser activation pattern is characterized by the pulse and line distances that are larger than the focus diameter. As it can be observed, this leads to the nucleation of Pd at the single spots corresponding to the laser-treated regions. Around these structures, some splashes (marked with arrows) from the laser process are visible. On the other hand, larger line distances, as well as shorter pulse distances (as shown in Figure 5b), lead to ablated lines in the scan direction. The local agglomeration of Pd also takes place in the regions affected by the laser treatment, and its concentration is significantly higher than in the non-ablated areas. Finally, shorter pulse and line distances were chosen in order to homogeneously treat the polymer surface (Figure 5c). As it can be seen, the palladium distribution achieves a relatively homogeneous distribution in both directions (scan and hatch directions).

However, also some non-activated areas with a lower content of Pd are visible in Figure 5c. This results from indirect modification with debris and activated particles originating from the interaction process. The interaction with the short laser pulses promotes photothermal and photochemical reactions at the surface and leads to the degradation and/or cross-linking of molecular chains and the amorphization of the surface [49]. These reactions can alter the polymer surface by incorporating new functional groups, which promotes the reduction of Pd(II) species to Pd [41]. Although the surface ablation is not completely homogeneous for the selected laser and scanning parameters, the presence of Pd was detected over the entire activation range. It can be assumed that the nucleation process is favored by direct laser activation and indirect modification (presence of deposits and spatter).

Besides this phenomenological effect, the amount of Pd was quantitatively determined from the EDX analyses in order to find explanations for the description of the process windows in the following subsection. Figure 6 shows the Pd content as a function of laser fluence from 7 to 35 J·cm^−2^ and pulse distances from 10 to 80 µm for PBT. The blue reference line shows only 0.03 wt.% Pd for a non-activated surface area after the nucleation process, which is below the detection limit of the EDX detector.

The elemental quantification of Pd, corresponding to the integral area, was investigated for different activation parameters (pulse and line distance). Figure 6a shows that the lowest selected laser fluences for the three parameter sets lead to the highest concentration of Pd in the measured area (e.g., from 0.57 to 0.94 wt.%). For higher fluences, the Pd content decreases (between 0.63 and 0.29 wt.%). By increasing the pulse distance (*d*_P_) from 10 to 20 µm, also a higher content of Pd was observed (see black squares and triangles in Figure 6a). It has to be mentioned that although higher laser fluences and shorter pulse and line distances resulted in inhomogeneous structures, these areas were also strongly burned (due to the increased energy input per area). This effect was already discussed in Section 3.1.

The effect of the pulse distance on the Pd content depending on the laser fluence was also investigated and is shown in Figure 6b. As it can be seen, the Pd content first increases with longer pulse distances and reaches its maximum at *d*_P_ = 40 µm (1.1 wt.%). After that, it decreases for longer pulse distances, and the surface topography changes from a homogeneous structure to a line-like or dot-like structure geometry. In addition, by comparing the palladium concentration with the roughness parameter *Rq*, it can be seen that with increasing roughness (Figure 4a,b), the Pd concentration decreases for all parameters shown in Figure 6. Furthermore, the highest Pd contents were determined for low *Rq* values (~1.70 µm).

The same analyses were performed for the PA material, as shown in Figure 7. The blue reference line shows 0.27 wt.% Pd for non-activated surface areas after the chemical baths. Figure 7a shows the Pd content as a function of laser fluence as well as pulse and line spacings. The highest Pd concentrations (between 5.42 and 22.04 wt.%) can be found at the pulse spacing of 40 µm for the small line spacings (10 and 20 µm). If the line spacing is increased from 10 to 20 µm, the average Pd contents increase. If the pulse distance is kept constant and only the line distance is increased to 80 µm, the Pd content strongly drops below 2 wt.%. The effect of the pulse distance can be seen in Figure 7b. The Pd content starts between 4.26 and 10.21 wt.% and decreases for longer pulse distances. Depending on the used line distance, the highest concentrations were also, in this case, measured at the lowest used laser fluences. A comparison to roughness shows that as roughness increases, Pd concentration drops as fluence increases. In addition to the laser fluence, the influence of the pulse distance is also considered; it can be seen that increasing pulse distance resulted in lower roughness and Pd content.

Comparing the two polymers, it is found that the initial concentration of Pd (PBT 0.03 wt.%) and the laser processed areas have a higher Pd concentration in PA. Furthermore, a general correlation can be recognized for laser fluence. With rising laser fluence, the roughness increases while the Pd content decreases.

It has to be mentioned that the absolute values of the mass fractions cannot be compared with each other since the elements and, thus, the masses differ. This leads to an overall lower concentration of Pd for PBT than for PA since the flame retardants Sb and Br are additionally included. Overall, the partially different behavior of the two polymers with respect to surface activation and the subsequent Pd concentration can be concluded.

### 3.3. Process Window Evaluation for Selective Metallization

In the third process step, the metallization of the treated polymers using chemical copper plating was conducted. The target is to produce homogeneous copper layers in the areas treated by the laser process with high electrical conductivity. Furthermore, it is also an attempt to achieve sharply contoured metallic structures with high contrast.

To evaluate the metallization result as objectively as possible, a routine using Matlab software was developed. This routine uses optical microscope images of the related fields (with dimensions of 8 × 8 mm^2^). Figure 8 shows exemplary the obtained results (see description of the procedure in Section 2.4), linking the generated geometric reference (Figure 8a) with the captured input image (Figure 8b), with several randomly selected fixed positions (blue dots). All pairs, e.g., 1 and 1’, are linked manually to find the correct alignment and orientation of the mask and the input image. The various possibilities of geometric transformation, such as scaling, rotation, tilting, and translation, result in the minimum of the required four-point pairs to realize the correct geometric mapping and alignment. Then, the algorithm calculates the number of pixels (marked in white in Figure 8c) of the target area that was covered with Cu, and the percentage value is calculated.

Using the above-mentioned percentage values, three conditions have been defined to describe the quality of the obtained copper coating for further evaluation. These conditions are (i) less than 2%: no copper; (ii) 2–90%: inhomogeneous layer; and (iii) more than 90%: homogeneous layer. As can be seen in Figure 8c, several laser process parameters lead to values below 2%, meaning that these regions were not covered with copper. Only 6 out of 25 process conditions permitted to obtain of and homogeneous Cu-layer, while eight parameter combinations produced an inhomogeneous layer.

The definition of these conditions permitted to represent the quality of the obtained Cu-films as a function of the pulse (*d*_P_) and line (*d*_L_) distances as well as the accumulated fluence (*F*_acc_). The last represents the total energy dose that is utilized to irradiate a certain area and, in this case, can be calculated by the used laser fluence (*F*) and the number of pulses (*N*) as follows:*F_acc_ = F × N*(3)

The obtained results are shown in Figure 9 for both PBT and PA materials.

The parameters for copper plating with values higher than 90% are marked by red symbols. These conditions determine the optimal parameters of the process. Significant differences can be seen for both materials with regard to the size of the experimentally determined process windows. Clearly, the selection of suitable accumulated laser fluences and associated scan parameters is much more critical for the PA. The optimal copper plating areas are followed by transition areas with inhomogeneous copper layers (blue symbols) up to non-metallized areas (black symbols). Differently, the PBT material has a wider range of possible parameters leading to homogeneous copper plating. In particular, there are a few combinations of parameters that can be implemented in both materials, which are *d*_L_ = 20 µm, *d*_P_ = 10 and 20 µm with a laser fluence of 7 and 14 J·cm^−2^, leading to accumulated laser fluences between 1,120,000 and 1,400,000 J·cm^−2^. Resistance is an important parameter for applications of the LSA method. The sheet resistances determined by a four-point measurement obtained are in the range of 0.10–0.22 Ω/sq and are comparable with previous literature [18,19]. The detailed evaluation of the sheet resistance will be the content of future publications.

## 4. Conclusions

In this study, the Laser-Based Selective Activation method was used for generating conductive structures in PA and PBT materials. Selected laser and scan parameters were used to target laser activation for the metallization of the polymer substrates, and their influence on the surface morphology, Pd content after the chemical bath, and, finally, on the quality of the Cu-layers was performed. Significant differences were found between the two studied polymers. Firstly, variation of the laser process conditions permitted to vary the surface roughness of the treated substrates up to ~5 µm and 3 µm for PBT and PA, respectively. Likewise, the influence of surface roughness on the palladium concentration could be shown. The highest Pd contents were reached for Rq values between 1 and 2 µm. However, the amount of Pd was up to one order of magnitude higher for PA compared to PBT due to the additives in the polymer matrix. With the aid of an image-based algorithm, the quality of the metallized structures was evaluated and compared to the used process parameters in order to determine the process window for successful Cu-deposition. In particular, the accumulated laser fluence was found to be a very important parameter. In the case of PBT, the optimal conditions correspond to pulse distances between 20 µm and 80 µm and accumulated fluences between 200,000 and 4,000,000 J∙cm^−2^. For PA, a full copper deposition is obtained using pulse distances between 20 µm and 60 µm and accumulated fluences from 1,000,000 to 4,000,000 J∙cm^−2^. The chosen approach, based on the LSA process, represents a good opportunity to selectively metallize other commercially available polymers without fillers and additives. The properties of the chemically deposited copper layers and their achievable contrast will be further investigated in future work.

## Figures and Tables

**Figure 1 materials-15-06572-f001:**
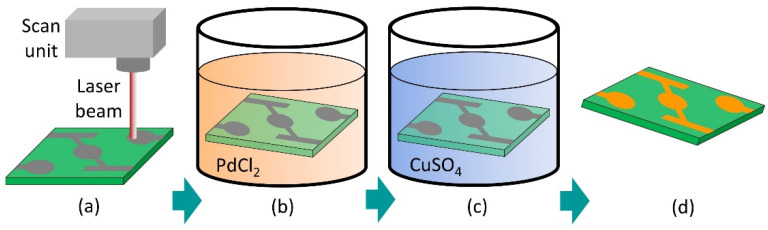
Schematic of the process chain for the fabrication of copper tracks. (**a**) laser activation, (**b**) infection bath, (**c**) selective metallization, (**d**) conductive copper tracks.

**Figure 2 materials-15-06572-f002:**
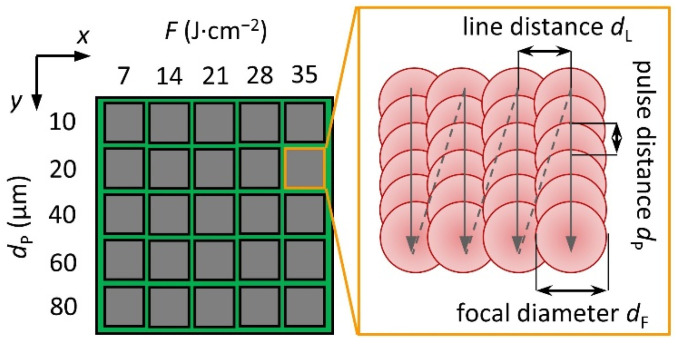
Schematic representation of laser and scanning parameters for the matrix (one red dot equals one laser spot; the gray arrows illustrate the scan path).

**Figure 3 materials-15-06572-f003:**
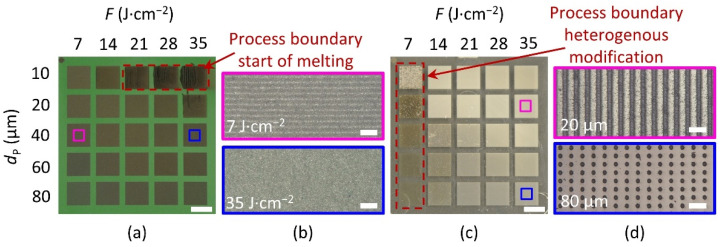
Representative photographs (**a**,**c**) and optical microscope (**b**,**d**) images of PBT and PA substrates denoting the resulting surface morphology after laser treatment. 5 × 5 matrixes in (**a**) PBT, with *d_L_* = 10 µm and (**c**) PA, with *d*_L_ = 80 µm. (**b**) Optical microscope image of the field treated in PBT with *d*_L_ = 10 µm, *d*_P_ = 40 µm and *F* = 7 and 35 J∙cm^−2^. (**d**) Optical microscope image of the field treated in PA with *d*_L_ = 80 µm, *d*_P_ = 20 and 80 µm and *F* = 35 J∙cm^−2^. Scale bars in (**a**,**c**) 8 mm; scale bars in (**b**,**d**) 100 µm.

**Figure 4 materials-15-06572-f004:**
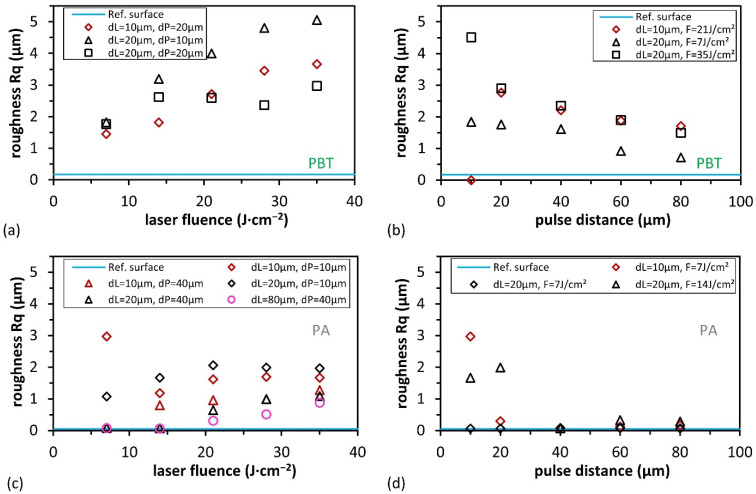
Surface roughness *Rq* for PBT and PA depending on (**a**,**c**) laser fluence and (**b**,**d**) pulse distance.

**Figure 5 materials-15-06572-f005:**
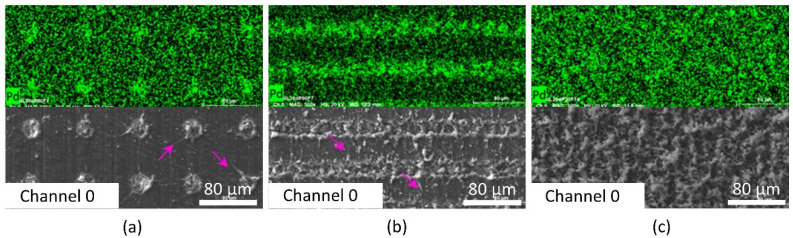
SEM image and EDX mapping of Pd (light green) for different scan and laser parameters of PBT. (**a**) *d*_L_ = 80 µm, *d*_P_ = 80 µm, *F* = 7 J∙cm^−2^, (**b**) *d*_L_ = 20 µm, *d*_P_ = 60 µm, *F* = 7 J∙cm^−2^ and (**c**) *d*_L_ = 20 µm, *d*_P_ = 20 µm, *F* = 14 J∙cm^−2^ (arrows (magenta) indicate splashes around the treated zones.

**Figure 6 materials-15-06572-f006:**
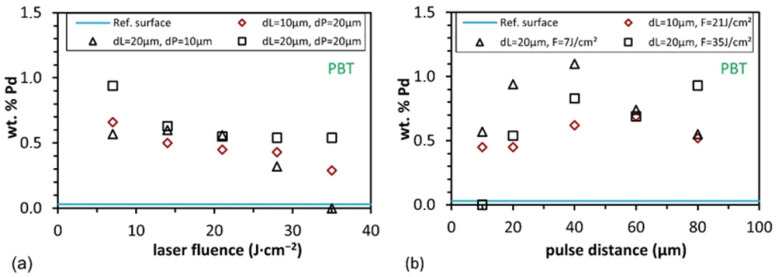
Measured Pd mass percentage (normalized) of PBT with different pulse and line distances and the non-activated reference surface (blue line) (**a**) depending on Laser fluence and (**b**) depending on pulse distance.

**Figure 7 materials-15-06572-f007:**
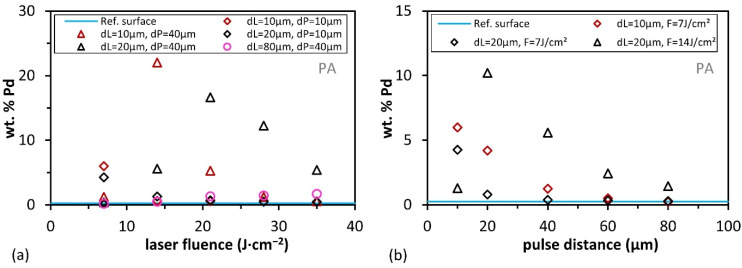
Measured Pd mass percentage (normalized) of PA with different pulse and line distances and the non-activated reference surface (blue line) (**a**) depending on laser fluence and (**b**) depending on pulse distance.

**Figure 8 materials-15-06572-f008:**
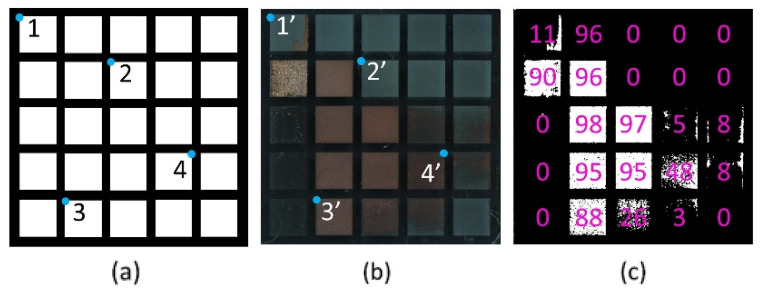
Schematic workflow to analyze the copper percentage (**a**) geometrical reference mask, (**b**) input image, and (**c**) binarized result with percentage of copper.

**Figure 9 materials-15-06572-f009:**
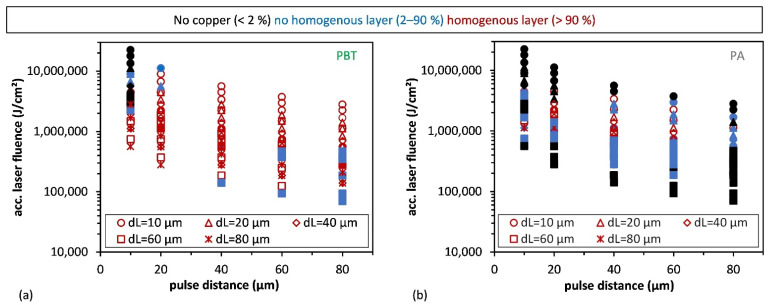
Calculated process window for coppering of PBT (**a**) and PA (**b**) depending on the accumulated laser fluence and the scanning parameters.

**Table 1 materials-15-06572-t001:** Laser-based methods to fabricate conductive tracks on dielectric materials.

Method	Material	Filler	Catalyst/ Activator	Wavelength [nm]	Pulse Duration	References
-	SiO_2_	-	PdSO_4_	248, 308	pw	[24]
-	PI	-	PdSO_4_, Pd(CH_3_CO_2_)_2_	248, 308	pw	[25]
LMD	PI	-	Cu^2+^	510.6, 514	cw, quasi cw	[4,6]
LMD	Al_2_O_3_, SiC, sapphire, diamond	-	Cu^2+^	193, 248, 308, 510.6, 10,600	ns	[5]
LDW, LCLD	PI, PET	-	[Pd(NH_3_)_4_]^2+^ /HCOH, tartarate-complex solution of Cu^2+^	248, 308, 488	cw, ns	[8,9,26,27,28]
LIFT	Si, SiO_2_	Cu film		193	ns	[11,29,30]
* LISA	PC	-	PdCl_2_	1064, 340	-	[14]
* LDS	ABS	CuO, Cr_2_O_3_	-	1064	-	[31]
* LDS	PDMS	Cu_2_(OH)PO_4_ ATO	-	1064	pw	[32]
* LISA	PC/ABS	-	PdCl_2_ mixed with SeCl_2_ and AgNO_3_	1064, 532	ps	[33,34]
* LDS	PP	CNT		1064, 532	ns, ps	[33,34,35]
* SSAIL	PC/ABS, PA, PEEK, PET, PMMA, PPA, PVC	-	AgNO_3_	1064, 532	ns, ps	[18,19,20,21,22,23]

* Marked literature is most relevant for discussion with own work.

**Table 2 materials-15-06572-t002:** Selected properties of the applied polymers [43,44].

	PBT	PA6
Structural formula	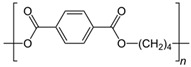	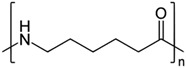
Oxygen bonding	Single and double bond	Double bond
Melting temperature	220 °C	220 °C
Additives	Sb, Br	-
Density	1.3 g/cm^3^	1.14 g/cm^3^
Thermal conductivity	0.27 W/K m	0.23 W/K m

## Data Availability

The data presented in this study are available on request from the corresponding author.

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
