# Peer review of "Modification of Polymeric Surfaces with Ultrashort Laser Pulses for the Selective Deposition of Homogeneous Metallic Conductive Layers"

_materials, 2022, doi:10.3390/ma15196572_

Round 1

Reviewer 1 Report

i would like to thank authors for their efforts .i have little comments then I recommend agree for publication review

 Review Report

I would like to thank all authors of the manuscript for their good and novelty manuscript titled as (Modification of polymeric surfaces with ultrashort laser pulses 2 for the selective deposition of homogeneous metallic conductive layers) which submitted to journal (Materials (ISSN 1996-1944).

1-The manuscript is original and novel as it aims to In this study, Laser Based Selective Activation method was used for generating con-435 ductive structures in PA and PBT materials. Selected laser and scan parameters were used 436 to target laser activation for the metallization of the polymer substrates, and their influ-437 ence on the surface morphology, Pd content after the chemical bath and finally on the 438 quality of the Cu-layers was performed2-The Presentation of the manuscript is good which attract the Interest to the readers. represents a good opportunity to selectively metallize other commer-453 cially available polymers without fillers and additives

3-Minor revision is needed to English language and style.

4-The introduction provide sufficient background and include all relevant references and styled according to the style of the journal.

5- All the cited references relevant to the research.

6- All the cited references relevant to the research.

7- The methods adequately described.

8- The results clearly presented.

9- The conclusions supported by the results.

So, I recommend accepting after minor revision (corrections to minor methodological errors and text editing.

Corrections are:

In  Introduction

Corrections are highlighted in the pages 1,2,3 lines(39,45,93)

In Materials and Methods

Corrections are highlighted in the page 4 lines(142,143)

In Results and Discussion

Corrections are highlighted in the page 8 lines(300)

Reviewer 2 Report

It is a very nicely presented piece of work and would be of significant interest to individuals involved in metallization and converting processes.

The following addresses the points highlighted:

1. The researchers investigated the effects of various process parameters on the metallization of two substrates PBT and PA6 with the objective to optimize the processing window and productivity. They try to solve the issue of flexible production of electrically conductive tracks for electronic circuits and systems.

2. The idea is quite relevant and interesting mainly for those involved in PCB manufacturing, metallization and converting in academia as well as in industry.

3. The idea is now new. It has been reported before, especially in a conference proceeding for PBT by the same authors but for PA6, it is new and in-depth.

4. No doubt, the paper is well written, and the text is clear and easy to read.

5. The conclusions are consistent with the evidence and arguments presented and surely the main question posed is addressed.  

However, it would have been nice if the authors could provide some information on the issue of heat accumulation and transient heat conduction induced by laser if any. Also, citation of some latest references would also improve the quality of the submitted manuscript.

Reviewer 3 Report

The article presents the results on the laser formation of conductive elements in a polymer. Technological stages are described, dependencies are given. The authors of the article have prepared a good material, however, it is worth discussing some fragments before publication:

The introduction provides a broad overview of laser processing techniques and the data are summarized in Table 1. However, this seems like a good literature review, as it is very difficult to understand the problem statement of this study. Of course, there is a problem, but the idea is spread too widely - from flexible technologies to photolithography. The reader may assume that everything has already been done in this area.

On page 2 (lines 48-54), the authors tried to generalize laser methods, but it turned out to be very concise and incomprehensible. The phrase " In all cases, the laser radiation is used to initiate or support the activation process on the technical surfaces. " is misleading, it can be understood ambiguously. It would be good to start from the general and go to the particular, that there are a number of physical mechanisms that can be triggered by methods.

It can be assumed that the authors propose to use laser radiation to generate certain metal clusters in a polymer matrix. However, direct laser structuring involves the use of other physical mechanisms of material processing, such as ablation, which can provide a change in the surface topography.

Sentence "Typical laser wavelengths range from the ultraviolet (UV) to the infrared (IR)." sounds unfinished.

Page 2 (line 58) - "by Shafeev et al." is without the reference.

Fragment of the paragraph "Laser Metal Deposition 61 (LMD) was investigated under air and resulted in catalytically active surfaces. The setup was also used for the treatment of PI with excimer lasers [4–6]." - sounds like a set of facts and it is difficult to understand whether this applies to the work, which the authors described at the beginning of the paragraph?

Each subsequent paragraph contains some fact that someone used the laser method, it does not evaluate the result obtained or information about the positive / negative aspects of this method.

The authors also touch upon the issue of 3D processing of materials, but it is difficult to evaluate this result in the review. Only in the LIFT it was indicated that 3D is impossible.

Sentence "The application of very short laser pulses" - confusing, is it about nonlinear processes and ultrashort laser pulses?

In the aim of the work, there is a quote "[40]", which may suggest that this has already been done earlier?

After such a relatively large review of the literature, it is very difficult to understand

1. Novelty of the work

2. Are the authors going to use 3D modification

Why does the roughness of the blanks ensure that palladium enrichment cannot take place on the substrate?

After the phrase "Previous experimental investigations have shown" readers can expect a link to this work.

It is not clear what are the coefficients A and PP in the expressions 1 and 2

Why did the authors choose the "pulse distance" parameter and not the overlap factor, as other researchers usually do?

Authors noticed colors in the processed regions. What color? and what was the reason for color formation? 

What is a pattern authors discussed in p. 7, lines 250-251?

What is the reason for the appearance of three classes of surface structures? 

The phrases such as "pulse distances" are difficult to understand.

Sometimes authors pointed the values of measured roughness with accuracy up to tenths and sometimes hundredths. What is the measurement accuracy? Please indicate the deviation interval on the graph. It'd be better to add it for all measured values.

Check please the construction of the sentence 382-384.

 "nonhomogeneous layer" sounds better than "no homogeneous layer"

In conclusion authors claimed "for generating conductive structures ". It'd be nice to see the results with resistance/conductivity measurement. Since the uniformity and thickness may influence on the results, it's expected to compare it to some similar published papers.

Annotation - I'd also recommend to update according to changes in the text.

Round 2

Reviewer 3 Report

The authors have reviewed and improved the manuscript. However, there are some issues to discuss:

- it's about overlapping of laser pulses. The maximum values can be 100% when we discuss the overlap. But how did authors obtain 166%? (line 184)

- please check Eq 1 and 2. All the variables have to be indicated in the text. What is RR?

- paper consists of numerous abbreviation that makes the iNTRO part hard to read. For example, line 44 presents "Laser-induced selective activation", but the line 123 demonstrates "laser-based selective activation". Is it really different techniques or just the same with different titles? Why?

- In general, the intro seems huge and complicated to follow. I'd recommend to optimize it.  

- Check abbreviations again, it seems that they repeat from paragraph to paragraph. For example, line 136 includes "PBT", and line 97 includes PBT.

- Fig 4 still requires deviation intervals. In the text we can also see values such as "2.97". Can the profilometer really measure with the specified accuracy? Or it is possible to specify only 3 microns value? There is no information about the resolution of a contact-type profilometer in section 2.

- Unusual title of the figure is noticed - "(Figure 3b, 7 J∙cm-2)"

- It's not the meaning of abbreviations, I belief. "content (EDX) and topography (SEM)" 

- "As it can be seen, the palladium distribution achieves a relatively homogeneous distribution in both directions (scan and hatch directions)."- Sounds controversial, especially in technical literature. At first glance, dark inclusions are visible in the light green region. I recommend reformulating the description of the results presented in Fig 5.

- Recently, I came across a work (https://doi.org/10.3390/nano12071127), where the authors managed to apply backside laser deposition of metal contacts. They are quite homogeneous and have a resistance of 0.6 Ω. Can the results be compared somehow?

- Try to improve the abstract since a the actuality sounds weak. Try to make it more clear. (it's a recomendation)
